# The Influence of Crystal Orientation on Subsurface Damage of Mono-Crystalline Silicon by Bound-Abrasive Grinding

**DOI:** 10.3390/mi12040365

**Published:** 2021-03-28

**Authors:** Wei Yang, Yaguo Li

**Affiliations:** 1School of Aerospace Engineering, Xiamen University, Xiamen 361005, China; yangwei@xmu.edu.cn; 2Fine Optical Engineering Research Center, Chengdu 610041, China

**Keywords:** diamond grinding, single crystal silicon, subsurface damage, crystal orientation

## Abstract

Subsurface damage (SSD) produced in a grinding process will affect the performance and operational duration of single-crystal silicon. In order to reduce the subsurface damage depth generated during the grinding process by adjusting the process parameters (added), experiments were designed to investigate the influence of machining factors on SSD. This included crystal orientation, diamond grit size in the grinding wheel, peripheral speed of the grinding wheel, and feeding with the intention to optimize the parameters affecting SSD. Compared with isotropic materials such as glass, we considered the impact of grinding along different crystal directions <100> and <110> on subsurface damage depth (added). The Magnetorheological Finishing (MRF) spot technique was used to detect the depth of SSD. The results showed that the depth of SSD in silicon increased with the size of diamond grit. SSD can be reduced by either increasing the peripheral speed of the grinding wheel or decreasing the feeding rate of the grinding wheel in the <100> crystal orientation, if the same size of diamond grit was employed. In addition, we proposed a modified model around surface roughness and subsurface crack depth, which considered plastic and brittle deformation mechanisms and material properties of different crystal orientations. When the surface roughness (R_Z_) exceeded the brittle-plastic transition’s critical value R_ZC_ (R_ZC<100>_ > 1.5 μm, R_ZC<110>_ > 0.8 μm), cracks appeared on the subsurface. The experimental results were consistent with the predicted model, which could be used to predict the subsurface cracks by measuring the surface roughness. However, the model only gives the approximate range of subsurface defects, such as dislocations. The morphology and precise depth of plastic deformation subsurface defects, such as dislocations generated in the fine grinding stage, needed to be inspected by transmission electron microscopy (TEM), which were further studied.

## 1. Introduction

Single-crystal silicon is widely used as a base material in solar cells, integrated circuits, and infrared optical systems. Silicon substrates are generally processed through cutting, grinding, thinning, and finally polishing. In the grinding process, bound-abrasive grinding has increasingly broad applications in the manufacturing of hard and brittle materials due to high efficiency in material removal and comparatively easy control of the surface figure [1]. Subsurface damage (SSD), which is mainly produced following the grinding process, must be removed in the subsequent processes such as Chemo-Mechanical Polishing (CMP). In silicon processing, SSD renders itself as amorphous layers, dislocations, subsurface cracks, etc. [2]. When cracks occur, the machining regime is referred to as “brittle mode machining”; if no crack appears, the machining mode will be in “ductile mode” [3]. Compared to the polishing process, grinding is more prone to brittle fracture and will induce cracks at the bottom of the subsurface. SSD will degrade the strength and reduce the lifetime of silicon substrate. Efforts have been made to suppress SSD induced from grinding, and to obtain a perfect surface.

Yan et al. [4] conducted diamond machining experiments on silicon specimens by using cutting tools with different rake angles and revealed that the SSD depths were increased with the increasing depths of the cut in grinding. Liu et al. [5] found that the change in regulation of the grinding-induced SSD was the same as the change in tendencies of the grinding force, and surface roughness. A number of SSD measurement techniques, such as the angled polishing method [6], ball dimpling [7], scanning infrared depolarization [8], cross-sectional transmission electron microscopy (TEM) [9], energy dispersive spectroscopy (EDS) [10], laser Raman spectroscopy technique [11], and the X-ray diffraction [12] method have been proposed.

The SSD should be removed in the subsequent processes. For these, SSD measurement techniques unavoidably changed or even destroyed the ground silicon surface, while nondestructive methods required high-performance measurement systems and could not be used in-situ. It is necessary to establish a mathematical model for measurement without destroying the sample. Many models have been proposed to evaluate the depth of subsurface cracks of brittle material caused by grinding in brittle mode. Lambropoulos et al. [13] established the relationship between the median crack depth and normal force of the optical glass based on the theory of fracture mechanics. Li et al. [14] established a relationship between surface roughness (SR) and SSD depth for optical materials based on the model established by Lambropoulos et al. Shen et al. [15] presented the relationship between the median crack depth and cutting depth for optical glass during the scratching process. Unlike isotropic materials like glass, single crystal silicon has anisotropy in the surface and exists through the process of brittle-ductile transition [16,17]. Zhang [18] developed an analytical model in the rotation grinding process to predict the SSD depth in the silicon wafer, which considered the effect of anisotropy in the grinding process. Li et al. [19] extended the model to the silicon obtained through the relationship between surface roughness and SSD depth by a CBN (Cubic Boron Nitride) grinding wheel. The SSD depth can be predicted by measuring surface roughness.

In this research, we carried out orthogonal experiments to investigate the SSD in the diamond wheel grinding of silicon. We detected the SSD through the Magnetorheological Finishing (MRF) spot technique, which measured the depth of SSD that was ground along <100> and <110> orientations. The experimental results indicated that the influence of process parameters, including crystal orientation, diamond grit size of the grinding wheels, and feeding rate on subsurface defects during the bound-grinding process, which could be used to reduce the SSD depth and improve the processing efficiency. Then, we proposed a modified model of the relationship between surface roughness (R_Z_) and SSD by extending Li’s model [14], which considered plastic and brittle deformation mechanism and material properties of different crystal orientations. The proposed model is expected to assess the subsurface damage depth by measuring the roughness of the surface (R_Z_) during grinding.

## 2. Experiments

### 2.1. Grinding Samples

Single-crystal silicon samples (n-type, (100) plane) with a diameter of 50 mm and 5 mm thick were employed in the experiments. All the samples fixed to a platform with a magnetic clapping device were ground on an ultra-precision grinding machine (Magerle, Switzerland), as shown in Figure 1. To reduce the number of trials, and by extension the experimental costs, the orthogonal experiments that had taken the effects of grain size, wheel feed rate, and wheel rotation speed into consideration were carried out as shown in Table 1 [20]. Two sets of the grinding trials’ feed directions were separately parallel to the surface crystallographic orientations of <100> and <110>, and the <110> orientations are at 45° angles to the <100> direction. To preclude possible subsurface damage induced before the trials, the grinding removal depth of all samples were both greater than 12 μm. The particular grinding conditions are listed in Table 2.

### 2.2. The Surface Roughness Measurement of the Ground Specimens

To get the relationship between SSD and roughness quantitatively, we examined the surface roughness along the <100> and <110> orientations using a contact profilometer Taylor Hobson 1250XL (Taylor Hobson, Leicester, UK), which were perpendicular to the grinding direction as shown Figure 2. The length of measurement and cut-off were according to ISO 4288–1996. Each sample was examined for three randomly selected positions, raw data of which are shown in Figure 3.

### 2.3. The Sub-Surface Damage Measurement of the Ground Specimens

SSD is rather difficult to directly observe and detect since it often exists beneath the ground surface at a certain depth. Many methods of detection have been developed, both destructive [2,21,22,23,24] and non-destructive [25,26], for the damage, such as the dislocation, amorphous and poly-crystalline layers, and other nano defects, which are often observed through the transmission electron microscopy (TEM) at a high resolution [27].

Angle polishing [28] and cross-sectional microscopy [2] are commonly used to detect micro subsurface damage like subsurface cracks (SSC), which initiate from the brittle removal mode. The MRF spot technique is used to measure SSCs in this paper, which will not introduce the additional damage [29], and is more efficient compared with angle polishing and cross-sectional microscopy. The silicon samples were spotted at three random positions along the radial direction with a commercial MRF machine (QED Technologies, Q22-400X, Rochester, NY, USA) and etching with “HNA” solution (HF (49%):HNO_3_ (70%):CH_3_COOH = 1:3:10) for 15 minutes at room temperature to make the subsurface cracks observable, as shown in Figure 4a. After that, the samples were flushed immediately with water, the ground surface and polished surface at MRF was imaged with an optical microscope (Leica-Camera, Leica DM4000M, Wetzlar, Hesse-Darmstadt, Germany), the grinding-induced SSCs were observed, and the horizontal distance between the last crack and the polished boundary at both edges were recorded, as shown in Figure 4b. Finally, using a profilometer (VEECO, VEECO Dektak 150, Plainview, NY, USA), the spot-depth profiles were measured across the centerline of a MRF spot, as shown in Figure 4c, and the SSCs depth was measured by applying the horizontal distance of the last cracks obtained from the microscope to the depth profile that yielded the SSD depth. The presented SSD has an average of three spot measurements.

## 3. The Modeling of Predicting SSD

Based on the previous experimental observations [30,31,32], the SSD system will be induced during an indenter loaded in silicon, as shown in Figure 5. When the normal indentation is small, the plastic deformed region accompanying the lateral cracks will be formed beneath the indenter. The median microcracks will emanate from the boundary plastic deformation zone, if the threshold for normal indentation for brittle-ductile transition is approached.

Lambropoulos derived an analytical model for median and lateral cracks depths based on micro indentation mechanics and a hill model for indentation of a sharp indenter [33]. Li imposed a correction factor on median crack depth, considering the effect of elastic stress field [14]. The following represents the depth of lateral and median cracks:(1)$$C_{li}=0.43{\left(\sin\psi \right)}^{\frac{1}{2}}{\left(\cot\psi \right)}^{\frac{1}{3}}{\left(\frac{E}{H}\right)}^{{}^{m}}{\left(\frac{P}{H}\right)}^{\frac{1}{2}}$$
(2)$$C_{mi}=(k\alpha {)}^{\frac{2}{3}}{\left(\frac{E}{H}\right)}^{\frac{2(1-m)}{3}}{\left(\cot\psi \right)}^{\frac{4}{9}}{\left(\frac{P}{K_{C}}\right)}^{\frac{2}{3}}$$
where *ψ* is the sharpness tip angle of the indenter, *K_C_* represents the fracture toughness of the workpiece, *E* is the elastic modulus, *m* is a dimensionless quantity ranging between 1/3 and 1/2, and
(3)$$\alpha =0.027+0.09(m-\frac{1}{3})$$

Gu calculated the area of contact projected in the normal direction and substituted the definition of hardness, the relationship between the median crack depth and penetration depth was expressed during the process of scratching [32]:
(4)$$C_{mi}=(k\alpha {)}^{\frac{2}{3}}\frac{{\left(E^{1-m}•H^{m}\right)}^{\frac{2}{3}}}{{\left({Κ}_{C}•\beta \right)}^{\frac{2}{3}}}\tan^{\frac{8}{9}}h_{i}{}^{\frac{4}{3}}=m_{1}h_{i}{}^{\frac{4}{3}}$$

The hardness was substituted into the lateral crack depth Equation (1):
(5)$$C_{li}=0.43{\left(\sin\psi \right)}^{\frac{1}{2}}\frac{E^{m}}{H^{m}\beta^{\frac{1}{2}}}{\left(\tan\psi \right)}^{\frac{2}{3}}h_{i}=m_{0}h_{i}$$
where $h_{i}$ is the grain penetration depth, *μ* is the depth ratio of removal depth to cutting depth, *β* represents the elastic recovery coefficient of the material.
(6)$$\beta =\frac{1}{4-3\mu +\mu^{2}}$$

The grinding process is similar to the process of a sharp indenter scratch test due to the same material removal mechanism as shown Figure 6. The size of plastic zone bi is equal to the depth of the lateral crack, which nucleate at the bottom of the ductile zone. Therefore,
(7)$$b_{i}\approx C_{li}-h_{r}=C_{li}-(1-\mu )h_{i}$$
the maximum peak height and valley depth of the ground surface roughness are between the ground surface and the bottom bi of the plastic zone [19].


(8)$$R_{Z}=\frac{\sum_{i=1}^{5}y_{pi}+\sum_{i=1}^{5}y_{vi}}{5}\approx C_{li}-(1-\mu )h_{i}\approx b_{i}$$


As shown in Figure 6, the depth of median cracks (SSCs) can be expressed:(9)$$SSD=C_{mi}-h_{r}=C_{mi}-(1-\mu )h_{i}$$

The relationship between *SSD* and *(R_Z_)* could be expressed by eliminating the penetration depth:(10)$$SSD=\frac{m_{1}R_{z}{}^{\frac{4}{3}}}{{\left(m_{0}-(1-\mu )\right)}^{\frac{4}{3}}}-\frac{(1-\mu )R_{z}}{m_{0}-(1-u)}$$

In the process of grinding, the influence of the anisotropy of mono-crystalline silicon (Table 3) and different types of ductile- and brittle-regimes on subsurface damage should be considered, the critical ground surface roughness *(R_Z_)* value for the ductile-brittle-transition to be expressed in Li’s model [19]:(11)$$SSD_{<100>}\left\{ \begin{array}{l} \approx b_{i}\approx R_{Z}\begin{array}{cc} & R_{Z}\leq R_{Z}{{}_{C}}_{<100>} \end{array}\\ =SSC_{<100>}=\frac{m_{1}{}_{<100>}R_{z}{}^{\frac{4}{3}}}{{\left(m_{0}{}_{<100>}-{\left(1-\mu \right)}_{<100>}\right)}^{\frac{4}{3}}}-\frac{{\left(1-\mu \right)}_{<100>}R_{z}}{m_{0}{}_{<100>}-{\left(1-u\right)}_{<100>}}\begin{array}{cc} & R_{Z}≻R_{Z}{{}_{C}}_{<100>} \end{array} \end{array}\right.$$
(12)$$SSD_{<100>}\left\{ \begin{array}{l} \approx b_{i}\approx R_{Z}\begin{array}{cc} & R_{Z}\leq R_{Z}{{}_{C}}_{<110>} \end{array}\\ =SSC_{<100>}=\frac{m_{1}{}_{<110>}R_{z}{}^{\frac{4}{3}}}{{\left(m_{0}{}_{<110>}-{\left(1-\mu \right)}_{<110>}\right)}^{\frac{4}{3}}}-\frac{{\left(1-\mu \right)}_{<110>}R_{z}}{m_{0}{}_{<110>}-{\left(1-u\right)}_{<110>}}\begin{array}{cc} & R_{Z}≻R_{Z}{{}_{C}}_{<110>} \end{array} \end{array}\right.$$

Therefore, the depth of *SSD* can be estimated by Equations (11) and (12).

## 4. The Results and Discussion

### 4.1. Morphology of Subsurface Damage

As presented in Section 2.3, the processed ground sample that was placed on the stage of an optical microscope (Leica-Camera, Leica DM4000M, Wetzlar, Hesse-Darmstadt, Germany), we moved the stage along the center line of the “D-shaped” spot. The morphology and distribution of subsurface damage induced by three different grinding wheels in some samples is arrayed as below (Figure 7). No cracks are observed in the subsurface ground by the D15A wheel, which is mainly removed by plasticity. The defects are regular grinding marks. Therefore, the position where the last wear scar disappears is defined during the ductile-regime mode of grinding. Obvious cracks appear on the subsurface of D91 and 80# grinding wheels, which are removed in brittle mode. The size of the cracks induced by the 80# wheel is larger than the D91 wheel. The subsurface defects are regular grinding marks by the D15A grinding wheel. For the coarse grinding, like the D91 and 80# wheels, the subsurface damage are obvious cracks.

### 4.2. Depth of Subsurface Damage

The R_Z_ values under different processing parameters were summarized in Table 2. Based on the model presented in Section 3, the sharpness angle of diamond grits varied in the range of 46°~71° [5]. The experimental and predicted SSD of all grinding samples is shown in Figure 8. Li [19] expressed the critical surface roughness R_Z_ value for the ductile-brittle-transition:(13)$$R_{ZC}=0.37•\frac{E}{H}•{\left(\frac{K_{C}}{H}\right)}^{2}$$

The (*R_Z_*) values of A1~A3 and B1~B3 were smaller than the *R_ZC_*, and no cracks occurred in the subsurface by the D15A wheel grinding along the <100> and <110> crystal orientations. Therefore, we used the grooves of grinding to represent the subsurface crack value, resulting in the experimental value being slightly larger than the theoretical value. Except for the D15A wheel, other experimental results were within the predicted range.

### 4.3. Effects of Crystal Orientation and Processing Parameters on SSD

When the abrasive grains are ground along the <100> and <110> directions, the subsurface cracks will extend downwards along the (100) and (110) planes and perpendicular to the subsurface, as shown in Figure 9. The Si–Si covalent bond density on the (100) plane is greater than the (110) plane. Therefore, the depth of SSC along the <110> orientation grinding is deeper than along the <100> orientation (see Figure 10), which agrees well with Gao’s findings.

In order to reveal the effects of the grinding parameters on the subsurface damage of mono-crystalline silicon grinding to guide processing for improving the processing productivity, a range of results of orthogonal experiments were analyzed. The influences of the grinding parameters, such as the grain size, wheel speed, and feed rate, were plotted in Figure 11. As can be seen from Figure 11, the depth of SSD for diamond grinding wheels deepens with the increase of grit size and feed rate. However, when the wheel speed is increased, SSD decreases.

## 5. Conclusions

In order to reduce the subsurface damage depth generated during the grinding process, by adjusting the process parameters, we carried out the orthogonal experiments including grinding wheel particle size, grinding wheel linear speed, and feed rate. Compared with isotropic materials such as glass, we considered the impact of grinding along different crystal directions <100> and <110> on sub-surface damage depth. The MRF spot technique for concurrently determining the depth of sub-surface damage and observing the morphology of subsurface damage at various depths was applied to bound-abrasive ground samples. We proposed a modified model of the relationship between surface roughness (R_Z_) and SSD, which considered plastic and brittle deformation mechanisms and the material properties of different crystal orientations. The proposed model is expected to assess the subsurface damage depth by measuring the roughness of the surface (R_Z_) during grinding. The following conclusions can be drawn:

Except for the D15A ground samples (surface roughness (R_Z_) < R_ZC_, R_Z<100>_ < 1.5 μm, R_Z<100>_ < 0.8 μm), where no cracks were observed on the subsurface, all the experimental results were within the theoretical prediction range. The relationship between SSD and surface roughness R_Z_ was shown to be a proportional function and in good accordance with the proposed model.Grinding experiments showed that the subsurface damage depth in samples ground along the <110> crystal orientation was larger than that along the <100> crystal orientation in the same processing parameters, since the Si–Si covalent bond density on (100) plane is greater than (110) plane.Whether it was grinding along the <100> or <110> direction, the trend of SSD changes with process parameters is shown as the depth of SSD increasing with increasing grit size and feed rate, which decreases with the wheel speed.

In summary, this paper proposed an anisotropic SSD model for monocrystalline silicon based on the SSD model of optical glass. Then, the MRF spot method was applied to measure SSD along the <100> and <110> orientations, which is in good accordance with the model. It could guide the next process like the CMP to remove the quantitative depth. This article does not have a quantitative study on the depth of plastic deformation subsurface defects caused by the D15A grinding wheel grinding, which needs to be further studied by the TEM method.

## Figures and Tables

**Figure 1 micromachines-12-00365-f001:**
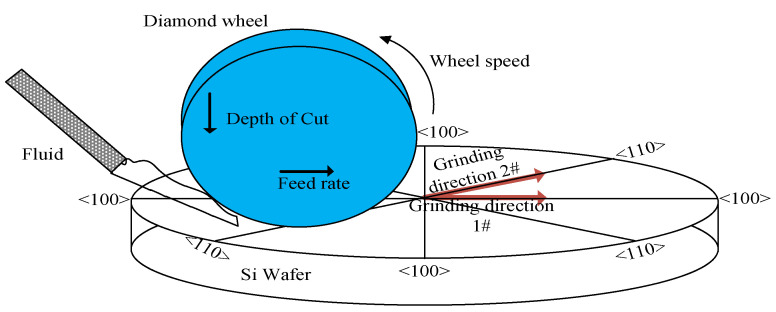
Schematic of trials of grinding along <100> and <110> orientation.

**Figure 2 micromachines-12-00365-f002:**
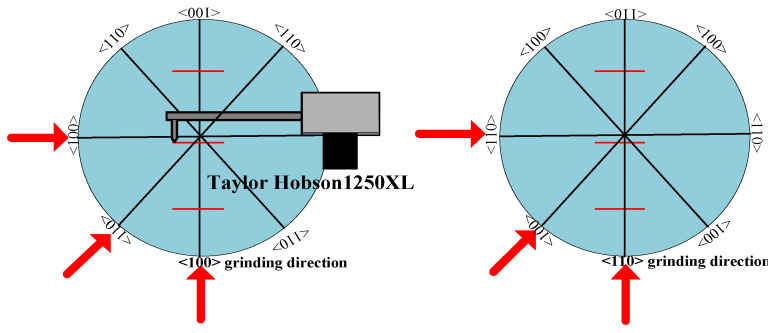
Measurement of the surface roughness of ground samples.

**Figure 3 micromachines-12-00365-f003:**
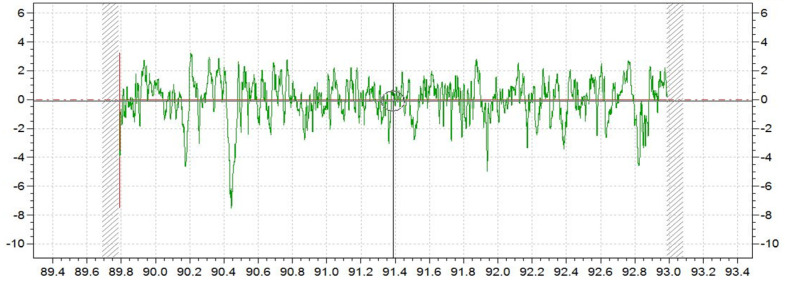
Surface roughness (R_Z_) of the grinding surface.

**Figure 4 micromachines-12-00365-f004:**
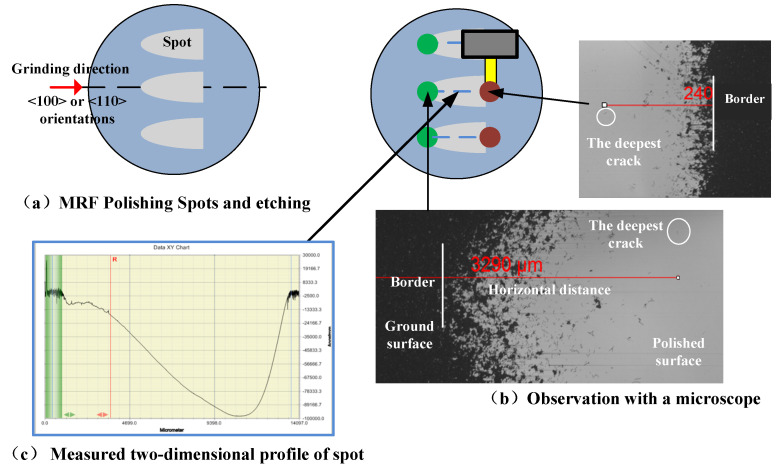
The sub-surface damage (SSD) measurement by spot magnetorheological finishing technique. (**a**) MRF Polishing spots and etching. (**b**) Observation with a microscope. (**c**) Measured two-dimensional profile of spot.

**Figure 5 micromachines-12-00365-f005:**
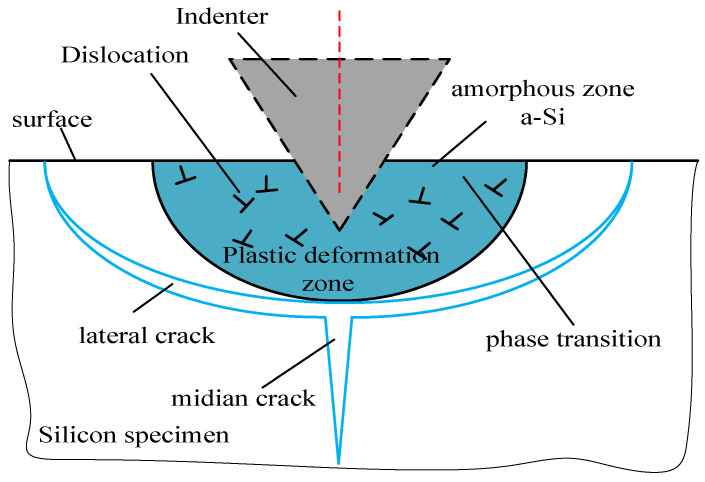
Schematic of damage by a sharp indenter.

**Figure 6 micromachines-12-00365-f006:**
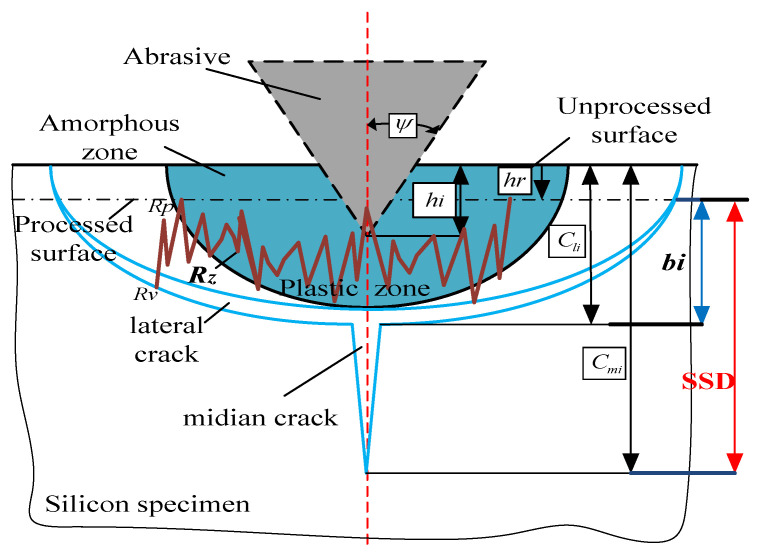
Schematic of subsurface-damage by an abrasive.

**Figure 7 micromachines-12-00365-f007:**
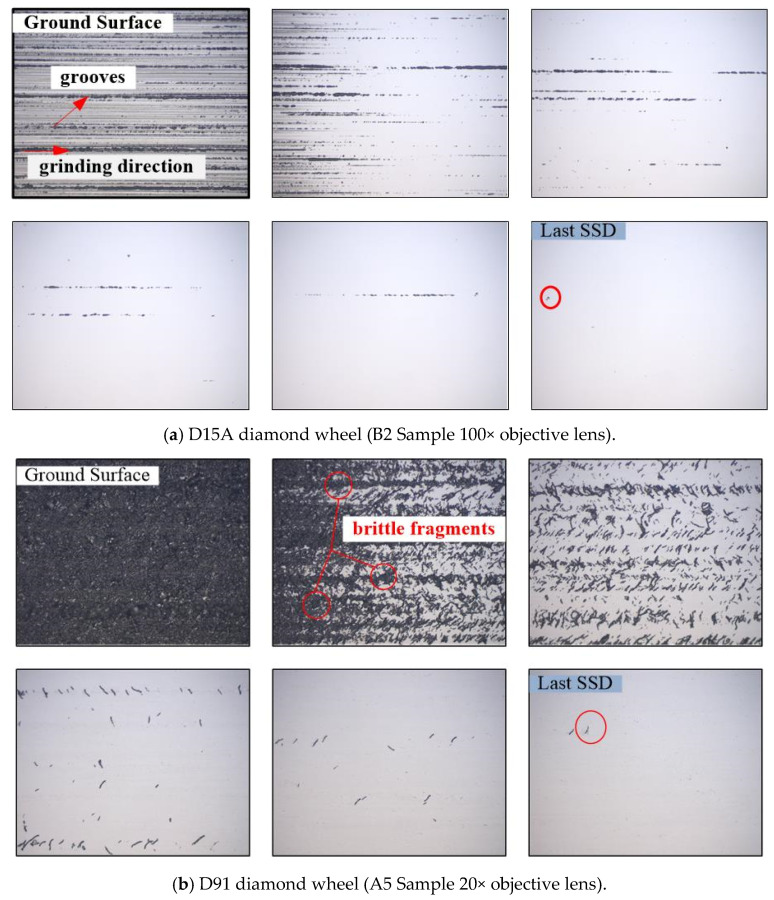
The morphology of subsurface in ground optical samples.

**Figure 8 micromachines-12-00365-f008:**
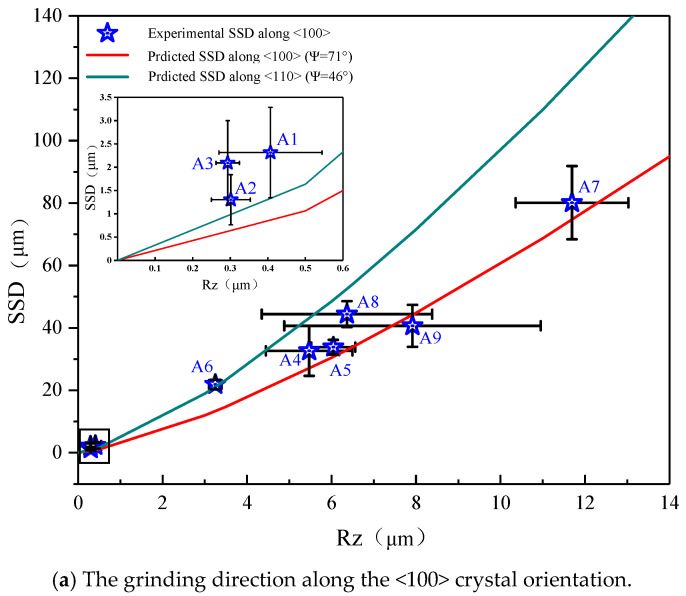
Comparison of experimental and predicted results.

**Figure 9 micromachines-12-00365-f009:**
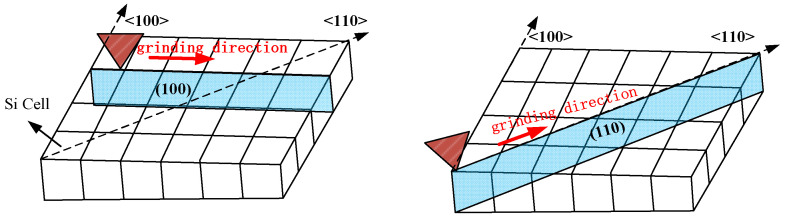
Schematic diagram of grinding direction and crystal orientation.

**Figure 10 micromachines-12-00365-f010:**
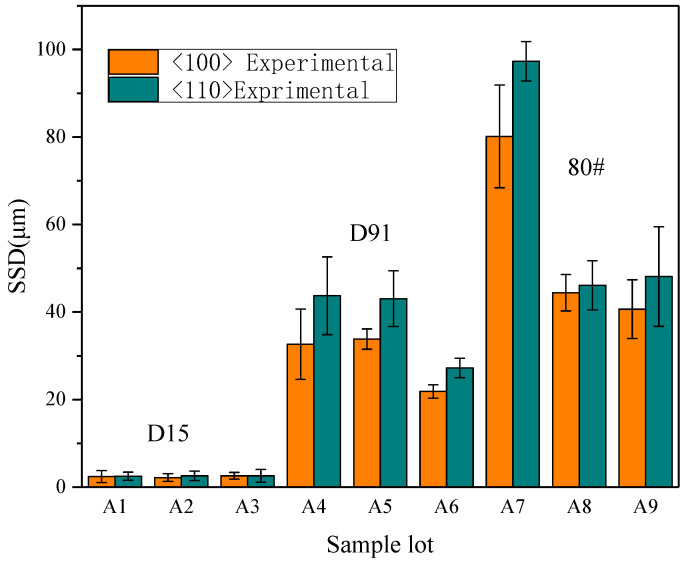
Comparison of the subsurface damage (SSD) along the <100> and <110> direction grinding.

**Figure 11 micromachines-12-00365-f011:**
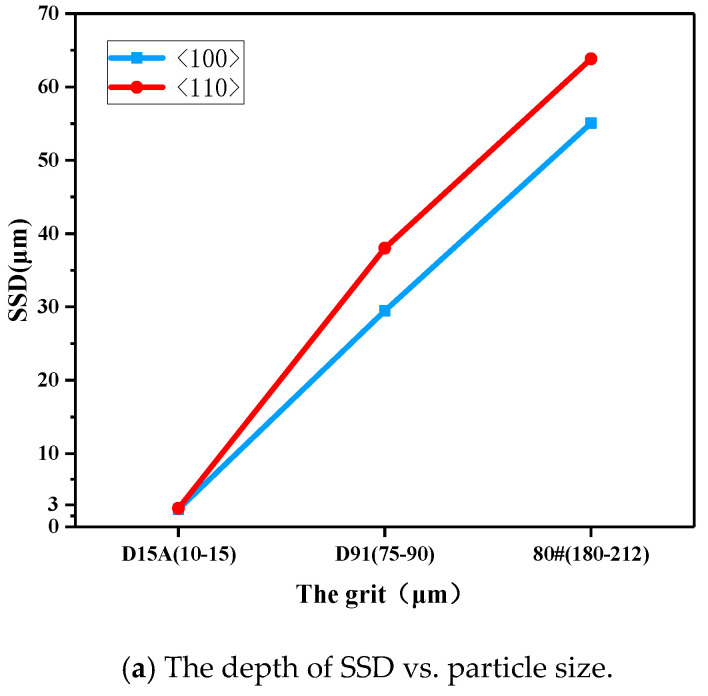
Sub-surface damage (SSD) vs. grinding parameters by range analysis: error bars represent standard deviations.

**Table 1 micromachines-12-00365-t001:** Experimental parameters and levels.

Factors	Parameters	Levels		
		1	2	3
A	Grain model [10] (μm)	D15A (10–15)	D91 (75–90)	80# (180–212)
B	Wheel speed (m/s)	10	20	40
C	Feed rate (mm/min)	300	1000	3000
D	Depth of cut (μm)	5	10	15

**Table 2 micromachines-12-00365-t002:** The experiment parameters and results (A, B represent the specimen along the <100> and <110> directions grinding).

NO.	Grain Model	Wheel Speed (m/s)	Feed Rate (mm/min)	Depth of Cut (μm)	<100> R_Z_ (μm)	<110> R_Z_ (μm)
A1 B1	D15A	10	300	5	0.4071	0.3914
A2 B2	D15A	20	1000	10	0.3061	0.3037
A3 B3	D15A	40	3000	15	0.2933	0.4486
A4 B4	D91	10	1000	15	5.4685	6.9261
A5 B5	D91	20	3000	5	6.043	6.9867
A6 B6	D91	40	300	10	3.2484	3.675
A7 B7	80#	10	3000	10	11.6924	12.9087
A8 B8	80#	20	300	15	6.3651	6.6319
A9 B9	80#	40	1000	5	7.9160	6.7404

**Table 3 micromachines-12-00365-t003:** The properties of the mono-crystalline silicon (N type, top surface is oriented in the (100) plane).

Crystalline Orientation	Hardness H (GPa) [33]	Elasticity Modulus E (GPa)	Fracture Toughness $K_{C}$ $\;(MPa\cdot m^{1/2})$	1-μ [8]	β
<100>	10	131	0.95	0.45	0.38
<110>		169	0.72	0.29	0.43

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
