# Peer review of "The Influence of Crystal Orientation on Subsurface Damage of Mono-Crystalline Silicon by Bound-Abrasive Grinding"

_micromachines, 2021, doi:10.3390/mi12040365_

Round 1
Reviewer 1 Report
Dear Authors
The reviewed article is very interesting. Below are some informative questions.
1. Figures 1 and 2 should contain the axes of symmetry.
2. What measure of scattering did the Authors used in Fig. 11?
Sincerely
Reviewer 2 Report
The reviewer comments of the paper «The Influence of Crystal Orientation on Subsurface Damage of Mono-crystalline Silicon by Bound-abrasive Grinding»- Reviewer
The authors presented an article «The Influence of Crystal Orientation on Subsurface Damage of Mono-crystalline Silicon by Bound-abrasive Grinding». However, there are several points in the article that require further explanation.
Comment 1:
The abstract needs to be completed. Demonstrate in the abstract novelty, practical significance. Add quantitative and qualitative work results to the abstract.
Comment 2:
Introduction
The introduction is well written. However, at the end of the introduction, add a clear and understandable purpose of the article.
Comment 3:
- Experiments
Why was Rz and not Ra assessed as a controlled parameter? What is the reason?
For measurement devices, software and machines used in research, indicate in parentheses (manufacturer, city, country).
Provide a more detailed description and explanation in the text of Figure 3. What is the reason for the microprofile pattern and what does it tell the reader?
Comment 4:
- The modeling of predicting SSD
Are all the formulas in the article original? If not needed appropriate citations.
Make sure that after each formula, the first used physical parameter is described and decoded.
Are all the figures in the article original? If not needed appropriate citations and publisher permissions.
Comment 5:
- Results and discussion
Describe in more detail in the text of the figures 7, 8, 9, 10, 11.
Comment 6:
It will be useful to add a section of Nomenclature in which to sign all the physical quantities and abbreviations encountered in the article. There are many physical quantities in the text and such a section will help to find the description of the necessary element.
For example,
ap : Depth of cut (um)
SSD : Sub-surface damage
etc.
Comment 7:
The conclusions need to be improved.
What is the novelty of the article? What is the practical significance? What are the differences from previous works?
Provide quantitative and qualitative conclusions for each parameter under study.
Conclusions should reflect the purpose of the article.
The article is interesting. However, the article needs to be improved. Authors should carefully study the comments and make improvements to the article step by step. All changes should be highlighted in color. After major changes can an article be considered for publication in the "Micromachines".
Author Response
Please see the attachment.
The front part of the attachment is the revised paper, and the back part is the reply to the reviewer.

Round 2
Reviewer 2 Report
The authors have improved the article according to the comments. The article can now be published.